# Sustainability Assessment of Combined Animal Fodder and Fuel Production from Microalgal Biomass

**DOI:** 10.3390/ijerph182111351

**Published:** 2021-10-28

**Authors:** Benjamin W. Portner, Antonio Valente, Sandy Guenther

**Affiliations:** 1Bauhaus Luftfahrt e.V., Willy-Messerschmitt-Str. 1, 82024 Taufkirchen, Germany; sandydguenther@gmail.com; 2Department of Chemistry and Applied Biosciences, Institute for Chemical and Bioengineering, ETH Zurich, 8093 Zurich, Switzerland; antonio.valente.g@gmail.com

**Keywords:** microalgae, biorefinery, fuel, fodder, feed, life cycle assessment, LCA, SLCA

## Abstract

We present a comparative environmental and social life cycle assessment (ELCA and SLCA) of algal fuel and fodder co-production (AF + fodder) versus algal fuel and energy co-production (AF + energy). Our ELCA results indicate that fodder co-production offers an advantage in the following categories: climate change (biogenic land use and land use change total), ecotoxicity, marine eutrophication, ionizing radiation, photochemical ozone creation, and land use. By contrast, the AF + energy system yields lower impacts in the other 11 out of 19 Environmental Footprint impact categories. Only AF + fodder offers greenhouse gas reduction compared to petroleum diesel (−25%). Our SLCA results indicate that AF + fodder yields lower impacts in the following categories: fair salaries, forced labor, gender wage gap, health expenditure, unemployment, and violation of employment laws and regulations. AF + energy performs favorably in the other three out of nine social indicators. We conclude that the choice of co-products has a strong influence on the sustainability of algal fuel production. Despite this, none of the compared systems are found to yield a consistent advantage in the environmental or social dimension. It is, therefore, not possible to recommend a co-production strategy without weighing environmental and social issues.

## 1. Introduction

Awareness of the detrimental impact that humanity has on the environment is growing worldwide and is becoming increasingly relevant in the public debate. In 2015, 195 countries adopted the Paris Agreement—the first globally binding covenant on climate—with the goal of limiting global warming well below +2 °C compared to the pre-industrial era [1]. It is well known that the massive use of fossil fuels is a driver of climate change. In 2018, the world primary energy demand amounted to 14.3 billion tonnes oil equivalent (Btoe), 81% of which were met by fossil fuels [2]. In the shift towards more sustainable energy sources, biofuels are expected to play a significant role [3]. Microalgal fuel, in particular, offers two advantages over first-generation fuels made from soybeans or rapeseed: Microalgae offer potentially higher biomass yields per unit area [4,5], and they can be grown on marginal lands, thereby avoiding competition with the food and fodder sector [6].

Despite these advantages, no commercial microalgal fuel plant exists today. Some authors explain this phenomenon by the high cost of production [7,8,9,10], whereas other authors have doubted the environmental benefits of algal fuel altogether [11]. In pursuit of a remedy, the concept of the algal biorefinery was born. Apart from oil, which is the raw material for fuel production, some algae species are capable of producing valuable co-products, such as cosmetic ingredients, pharmaceutical compounds, and pigments [12]. These could offer additional income and share part of the production burden. However, apart from the technical difficulties of recovering co-products in sufficient quantity and quality [12,13], not all co-production strategies are compatible with algal fuel. Laurens et al. [14] and Subhadra and Edwards [8] showed that the small market volumes typically associated with high value commodities, such as cosmetics, food supplements, and pharmaceuticals, are easily saturated by large-scale biorefineries. To avoid the uncertain socio-economic consequences of market glut, we prefer to combine algal fuel production with bulk co-products, such as chemicals and animal fodder. Several studies have investigated the suitability of algal biomass as a dietary supplement for poultry, pigs, ruminants, and aquaculture [13,15,16,17] with promising results. The observed benefits include improved overall health, better immune response, higher fertility, and increased body weight and product output [13]. Based on these findings and on the fact that a large part of today’s algal biomass production is already used for fodder [12,13], our study focuses on this co-product specifically.

Before algal fuel and fodder co-productions are employed at scale, it must undergo stringent sustainability assessments. Life cycle assessments (LCA) found in the literature [18,19,20,21,22] typically focus on greenhouse gas (GHG) emissions and disregard or underrate other issues. Social aspects of biofuel production, in particular, have demonstrated to be hard to quantify because of supply chain complexity [23]. Existing studies address bioelectricity [23], bioethanol [24,25], or biohydrogen [26]. Notable studies dealing with microalgal fuel in particular include Tavakoli and Barkdoll [27] and Rafiaani et al. [28]. However, none of the existing studies explore the effect of different co-production strategies on the social performance of microalgal fuel.

The goal of our study is to complement the existing literature by offering a broader view on the sustainability of microalgal fuel and fodder co-production. We present an environmental and social life cycle assessment (ELCA, SLCA) that includes all 19 indicators (the EF method consists of 16 indicators and 3 subindicators (climate change biogenic, climate change fossil, climate change land use, and land use change). Throughout our study, we will refer to them simply as 19 indicators for brevity) of the Environmental Footprint (EF) 2.0 method [29], as well as nine social indicators from the Product Social Impact Life Cycle Assessment (PSILCA) v3 database [30]. To the best of our knowledge, this is the first study to analyze both the environmental and social performance of microalgal fuel by using a comprehensive set of indicators. The reference for our comparison is an alternative algal fuel co-production pathway in which heat and electricity are produced from the residual biomass. We further compare the environmental impacts to those of petroleum diesel.

## 2. Materials and Methods

### 2.1. Goal Definition

The goal of our study is to quantify and compare the environmental and social life cycle performance of two algal fuel co-production systems. Both systems produce fuel via the hydrotreatment of algal oil (hydrotreated esters and fatty acids, HEFA) but differ in the utilization of non-oil biomass fractions. The AF + energy system converts residual biomass into electricity and heat via anaerobic digestion and biogas combustion (Figure 1a). The AF + fodder system converts residual biomass into animal fodder via spray-drying (Figure 1b). We perform an environmental life cycle assessment (ELCA) and social life cycle assessment (SLCA) for both systems in order to identify the more sustainable option. The highlighted environmental and social bottlenecks can further guide future development.

### 2.2. System Description

The process chain from algae cultivation to fuel final use is identical for both systems and is briefly recapitulated in Section 2.2.1. Note that this part of the system has been adapted from Portner et al. [31], and we refer to that study for an in-depth description of modeling assumptions and parameters. The central focus of this study—the co-production of energy and fodder—is described in Section 2.2.2. All models used in this study are available for download free of charge: the AF + fodder model from the Appendix A of this paper and the AF + energy model from the supporting information of Portner et al. [31]. Note that several adaptations were made to the AF + energy model, which are also described in the Appendix A of this study. The summarized bill of materials for both systems is given in Table 1 and Table 2.

#### 2.2.1. Algal Fuel Production

The values and assumptions stated throughout this subsection are taken from [31] and explained therein.

Microalgae are cultivated in open raceway ponds (ORP) in a coastal area in Spain. The ponds are excavated from the ground and covered by plastic liners. The cultivation mode is autotrophic, meaning the microalgae thrive on photosynthesis. ${\mathrm{CO}}_{2}$ is pumped into the pond after being extracted from the flue gas of a combined heat and power plant (CHP), which uses biogas as its primary fuel. The electricity and heat produced by the CHP are treated as co-products of the fuel production process. Nitrogen and phosphorus are supplied in the form of commercial urea and triple superphosphate (TSP) fertilizers. To ensure homogeneous exposure to the sun, the pond is mixed by paddle wheels throughout the day (12 hours per day). The power demand for this is estimated to be 4000 W/ha, or 30 GJ per day for the entire cultivation area of 175 ha. The biomass yield is estimated to be 15 g${}_{DW}$/(m² d) or 6.3 kilotons during the cultivation season of 240 days per year. The algae cells are subjected to nitrogen-stress at the end of their growth cycle to raise the lipid content to 30% by weight. The cells also contain hydrocarbons and proteins, which are valuable nutrients for animals. Water evaporates continuously from the open ponds (1.3 m³/(m² d)) and needs to be replenished by pumping saltwater from the neighboring sea via a dedicated pipeline. The power demand for pumping is estimated to be 160 kW or 6.9 GJ per day. Thus, the total power demand for algae cultivation amounts to 37 GJ per day. By contrast, the cultivation’s energy output in the form of algal biomass is roughly 630 GJ per day (assuming a mean lower heating value of 24 MJ per kg). We assume that the algae cells can tolerate salt concentrations up to 5.3%-wt and that salt accumulation beyond this point is prevented by reducing the recycling rate. In this manner, no external freshwater source is necessary.

After reaching the targeted cultivation density (0.5 g${}_{DW}$/L) and cell lipid content (30% by weight), the microalgae are harvested in a two-step procedure: First, the medium is pre-concentrated by flocculation with magnesium hydroxide. After that, it is centrifuged and yields a biomass concentration of 20% by weight. HCl is subsequently added to the centrate and to the supernatant in order to neutralize pH and to recover the flocculation agent. If salt levels permit, the neutralized supernatant is then returned to the cultivation process. Otherwise, it is discarded. The discarded medium is treated in a wastewater treatment plant and returned to the sea. Note that the composition of marine cultivation media after nitrogen deprivation and harvest is non-existent in the literature. For this reason, we chose a generic treatment model from the Ecoinvent database *treatment of wastewater, average, capacity 1.1E10l/year* [32].

After the centrifuge, the algae cells pass through a mill, which breaks open the cell walls. The released lipids are then extracted using hexane, and the extracted oil is shipped to the Netherlands where it is converted into fuel via hydrotreatment (hydrotreated esters and fatty acids, HEFA). Finally, the fuel is distributed across Europe to its final users. Combustion emissions are treated as carbon neutral, as the released ${\mathrm{CO}}_{2}$ is of biogenic origin.

#### 2.2.2. Co-Production of Energy and Fodder

Apart from algal oil, the extraction process produces an aqueous residue, which is rich in carbohydrates and proteins. This residue can be utilized in various ways. In our study, we compare two utilization scenarios: energy production and fodder production.

In the AF + fodder scenario, the residue is dried and sold as animal fodder. Not all drying processes are suitable for this task. The goal is to produce a durable product while conserving the bio-functionality of the algae proteins. On an industrial scale, spray-drying is employed for the production of baby formula and of Spirulina powder for food supplements [12,32]. We accordingly chose Ecoinvent activity *milk spray-drying, CA-QC* [32] as the basis of our model. We adapted the model to account for the different water content of the raw material (85% instead of 50%) and for the changed location (inputs from Spanish markets instead of Canadian ones). Furthermore, we added an efficiency term to model a loss rate of 10% of the nutritional value of the raw input. We assume that the remaining 90% of algal residues can displace soy meal on a 1:1 basis. Since no Spanish soy meal market was available from the Ecoinvent database, we approximated it to consist of 55% imports from Brazil and 45% imports from the United States (US), based on data from UN COMTRADE [33]. Due to the high water content of the raw input, spray-drying requires a significant heat input, which is satisfied by a natural gas CHP. Overall, the AF + fodder system is a net heat *consumer*. In accordance with the Environmental Footprint method [29], we assigned no dissipation impact to water evaporated from the algal residues, as it is mostly seawater. The resulting net inventory for the AF + fodder system, including fuel and fodder production, is reported in Table 2.

In the AF + energy scenario, the oil-extracted residue is used to produce biogas via anaerobic digestion (AD), which is then supplied to the biogas CHP to produce heat and electricity. Apart from biogas, AD produces digestate, which is rich in carbon, nitrogen, and phosphorous and can be used as a nutrient source for algae cultivation. Our model is based on Portner et al. [31] with adaptations described in the Appendix A. In summary, the effect of AD is fourfold: (1) algae-derived biogas reduces market biogas demand (−13% compared to system without AD); (2) nitrogen in the recycled digestate reduces market urea demand (−45%); (3) phosphorus in the recycled digestate reduces market TSP demand (−45%); and (4) carbon in the recycled digestate reduces cultivation-${\mathrm{CO}}_{2}$ demand (−16%). The reduced ${\mathrm{CO}}_{2}$ demand has further consequences for the biogas-CHP plant: As less ${\mathrm{CO}}_{2}$ is consumed, less electricity and heat can be attributed to the algal fuel as by-products (−16% each). Furthermore, the reduced ${\mathrm{CO}}_{2}$ demand results in reduced market biogas consumption (−16% on top of the reduction induced by local biogas production). The resulting net inventory of the AF + energy system, including fuel and energy production, is reported in Table 1. Note that electricity production by the AF + energy system is *lower* than in the AF + fodder system due to the described effects of AD.

### 2.3. Assessment Methodology

#### 2.3.1. Environmental Life Cycle Impacts

Assessment of the environmental performance is based on the standardized Life Cycle Assessment (LCA) methodology [34,35]. LCA was chosen because it allows a) evaluating the impact of human activities on different areas of protection, b) identifying environmental hotspots in the supply chain, and c) highlighting burden-shifts between different areas of protection and life cycle stages. For these reasons, it is also becoming an essential tool to underpin evidence-based policies in the EU [36]. According to the ISO standards, LCA comprises four interrelated stages [34,35].

In the first stage (“goal and scope definition”), key aspects such as the functional unit (FU) and the boundaries of the product system are defined. Since the main function of the systems under study is fuel production, we selected 1 MJ of fuel (lower heating value) as the functional unit. Concerning the system boundary, a cradle-to-grave approach was followed, covering feedstock production (including infrastructure), feedstock preparation, conversion, fuel distribution, and fuel combustion (Figure 1a,b). The geographical scope of the foreground system comprises Spain (algal oil production) and the Netherlands (fuel production). The temporal scope is present time, i.e., we consider state-of-the-art technologies.

The second stage (“life cycle inventory analysis”, LCI) focuses on the acquisition of input and output data (bill of materials) describing the production system. Additionally, an approach to deal with multifunctionality has to be detailed if the system under study produces more than one useful product. The bills of materials for the AF + energy system and the AF + fodder system are given in Table 1 and Table 2, respectively. The systems were modeled in Excel, and a workbook containing the AF + fodder model is available in the Appendix A of this paper. The AF + energy model is available from the supporting information of Portner et al. [31]. Modifications to the latter model were necessary and are also described in the Appendix A of this paper. The background system in our study is modeled using activities from the Ecoinvent 3.7.1 APOS database [32]. Multifunctionality in the foreground system is resolved following the substitution approach, in accordance with the ISO recommendation [34,35]. The two systems produce three useful products each (AF + energy: algal fuel, electricity, and useful heat; AF + fodder: algal fuel, electricity, and animal fodder). It is assumed that electricity displaces the current average Spanish grid mix, heat displaces heat generation from natural gas, and fodder displaces soybean meal. As no soybean meal market for Spain was available from the Ecoinvent database, we approximated it to be 55% Brazilian imports and 45% US imports, based on data from UN COMTRADE [33]. We assume that 1 kg of spray-dried algae-residues are nutritionally equivalent to 1 kg of soybean meal.

The third stage of LCA (“life cycle impact assessment”, LCIA) includes three mandatory components: (i) selection of impact categories, indicators and characterization models, (ii) linking of impact categories and inventory data (associating elementary flows with impact categories), and (iii) characterization of impacts (applying indicator-specific intensity variables, the characterization factors). The optional normalization and weighting step is omitted in our study. Given the European context, environmental life cycle performance was characterized by using the Environmental Footprint 2.0 method (EF 2.0) midpoint indicators [29]. We used all 19 indicators to capture a broad view of the possible environmental consequences and trade-offs. Linking and characterization (components ii and iii) are performed in Brightway 2 [37].

The last stage of LCA (“interpretation”) summarizes the findings of the LCI and LCIA stages, identifies critical life cycle phases, and gives recommendations for future development. This stage corresponds to Section 3 of this article.

#### 2.3.2. Social Life Cycle Impacts

The SLCA methodology is defined in the UNEP/SETAC Guidelines for Social Life Cycle Assessment [38] and is largely analogous to the LCA framework. It provides a framework for the quantification of social risks along supply chains, namely for the identification of hotspots in the social conditions under which a product and its components are produced. The social life cycle inventory is defined in terms of workhours per functional unit and characterization factors describe the risk of a specific social issue occurring in a country-specific sector. Inventories can be defined explicitly by the SLCA practitioner, or they can be taken from economic databases. We chose the first approach for the foreground system and the second approach for the background system.

For the foreground inventory, the source country of each exchange was first identified based on global commodity trade statistics reported in the UN COMTRATE database [33]. In a second step, the sector associated with each flow was selected among those available in the PSILCA 3.0 database [30]. Physical exchanges were converted to monetary units based on price data from the Ecoinvent APOS 3.7.1 database [32]. Prices in EUR were converted to USD using a conversion factor of 1.179 EUR/USD. The labor demand at the algal plant was estimated to be 240 days per year, 12 hours per day, and 10 workers per shift. When normalized by the fuel production rate (5.15·10⁷ MJ per year), this yields a labor demand of 5.59 × 10⁻⁴ work-hours per MJ fuel.

Linking and characterization was carried out in OpenLCA [39]. The PSILCA v3.0 database [30] was used to model sector and country specific background inventories, as well as their social risk levels. There are 55 social performance indicators available in PSILCA 3. We selected a subset of nine based on the following considerations: (i) relevance to central subjects of the SDGs; (ii) recommendations set by previous SLCA studies on alternative fuels in the European and Spanish context [26,40,41,42]; and (iii) socio-economic specifics of countries involved in the supply chain. The nine chosen indicators are as follows: child labor (CL); contribution of the sector to economic development (SED); fair salary (FS); frequency of forced labor (FL); gender wage gap (GWG); health expenditure (HE); unemployment (U); violations of employment laws and regulations (VEL); and women in the sectoral labor force (WLF).

## 3. Results and Discussion

### 3.1. Environmental Life Cycle Impacts

Figure 2 compares the calculated environmental impacts of the AF + energy system (blue) and the AF + fodder system (orange). The AF + energy system performs better in 11 out of the 19 indicators (climate change—fossil; ecosystem quality—acidification, freshwater and terrestrial eutrophication; human health—carcinogenic and non-carcinogenic effects, ozone depletion, and respiratory effect; resource depletion—water dissipation, fossil, materials, and metals). In the remaining eight categories, the AF + fodder pathway shows lower impact (climate change—biogenic land use and land use change total; ecosystem quality—freshwater ecotoxicity and marine eutrophication; human health—ionizing radiation and photochemical ozone creation; resource depletion—land use). Thus, we find no systematic environmental advantage for either co-production strategy.

Figure 2 further shows the environmental impact of petroleum diesel (green), which outperforms both algal fuels in 14 out of the 19 indicators. To understand this result, one must first understand the origins of impacts in the algal fuel systems. Two major contributors are the biogas CHP plant and the wastewater treatment plant (Figure 3). The CHP plant’s impact is explained by the resource-intensity of biomass production and the emissions created during biogas production and combustion (detailed in Section 3.1.1, Section 3.1.2, Section 3.1.3 and Section 3.1.4). Although the co-production of electricity and heat can compensate these burdens in four of the EF 2.0 categories, the CHP plant presents a net positive (i.e., damaging) contribution in the other 15 categories. Employing the presented algal fuel production system at scale would, thus, result in burden-shifting—i.e., the improvement of one impact category (e.g., climate change) at the cost of others. One way to prevent this is to explore different $CO_{2}$ sources for algae cultivation. For example, instead of digesting organic compounds in wastewater to produce biogas, which is then combusted to produce $CO_{2}$ and is then used to grow algae, it is more efficient—and likely environmentally beneficial—to grow microalgae directly in the wastewater in a heterotrophic mode. Concerning the wastewater treatment, its impact is influenced by two factors: the amount of effluent and its pollution. Although the amount can be minimized by recycling the cultivation medium after harvesting, there are limits imposed by the tolerable salinity and the accumulation of biotoxins (cf. [31]). The pollution can be limited by operational adjustments, which ensure that carbon-rich, nitrogen-rich, and phosphorous-rich compounds are mostly assimilated by the time of harvest. The authors are not aware of any existing studies that investigate the composition of spent algae cultivation media and/or the influence of operating conditions in the cultivation phase. Thus, this finding remains to be validated by experimental evidence. Ideally, such experiments would conclude that the cultivation effluent contains minor pollutants and that it can be diverted into natural water bodies without additional treatment.

Our results are supported by the existing literature, although differences in modeling assumptions and methodology complicate direct comparisons. Soh et al. [19] carried out growth experiments on different microalgae species and performed an LCA for a hypothetical biodiesel plant with protein and energy co-production. Our calculated climate and marine eutrophication impacts are within the range of their results Soh et al. [19] and Pérez-López et al. [21] report eutrophication as an aggregated score as per TRACI 2.0 method. In contrast, eutrophication in EF 2.0 is disaggregated into three subcategories: marine, freshwater, and terrestrial. Here, we compare the TRACI aggregate to the EF marine score. By analyzing a similar product system, Pérez-López et al. [21] performed an extensive uncertainty analysis. Whereas our AF + fodder GHG score lies between the 25% percentile and the median, our marine eutrophication score lies beyond the 95% percentile. This discrepancy stems from the system model: Wastewater treatment, which is the main contributor to marine eutrophication in our study, was neglected. Lastly, Gnansounou and Kenthorai Raman [20] compared algal biodiesel and -protein to petroleum diesel and soy, finding a moderate GHG reduction of −24%, which is identical to our AF + fodder scenario. They further support our finding that the fossil depletion score is on the same order as that of the fossil reference. Unfortunately, land use cannot be compared because different characterization metrics have been used (m² vs. points). Generally, no comparable LCA studies on algal fuel and fodder co-production could be found that analyzed impact categories other than climate change, eutrophication, fossil resource depletion, and land use.

In the following, we explore the origin of impacts within the algal fuel production systems in more detail, breaking them down according to process step and impact category (Figure 3).

#### 3.1.1. Climate Change

The climate change subcategory comprises the aggregated indicator "*climate change—total*" (CC total), as well as three subcompartments for biogenic methane emissions (CC bio), fossil GHG emissions (CC fossil), and land use change effects (CC LUC).

AF + fodder shows the lowest total climate change impact thanks to the credits for the displacement of soybean meal. The displacement of soybean cultivation in Brazil, in particular, yields a significant credit in the CC LUC subcategory. Both algal fuel pathways further profit from the displacement of fossil electricity (CC fossil). Despite these substantial credits, the AF + fodder fuel achieves only 24% GHG reduction compared to petroleum diesel, which is insufficient for RED II accreditation. The GHG intensity of the AF + energy fuel surpasses that of petroleum diesel.

Impacts of both algal fuel systems stem mainly from the consumption of electricity, urea fertilizer, hydrochloric acid, and from the treatment of discarded cultivation medium. Note that both algal fuel systems are net electricity *producers*. Subtracting the CHP credit (blue bar) from the electricity burden (dark orange bar) yields a net credit. Similarly, urea consumption is partially offset by digestate recycled from the anaerobic digestion process (violet bar). We decided to show both sides of the balance for transparency. Note that the release of ${\mathrm{CO}}_{2}$ from the cultivation ponds into the atmosphere has no impact, as the ${\mathrm{CO}}_{2}$ is of biogenic origin. Biogenic methane leaking from the anaerobic digestion (AD) process, on the other hand, causes a climate impact and is accounted for in the "*anaerobic digestion—total*" credit.

#### 3.1.2. Ecosystem Quality

Both algal fuels show substantially higher ecosystem quality (EQ) impacts than petroleum diesel in all five subcategories. These impacts stem mainly from the treatment of discarded cultivation medium and from the CHP supply chain. The former causes eutrophication by releasing nitrogen-rich and phosphorous-rich compounds into water and the atmosphere (EQ FW, EQ mar, and EQ terr). The CHP contributions are caused by burning digester sludge, which is a by-product of biogas generation. Acidification impacts (EQ acid) are governed by the consumption of grid electricity (${\mathrm{SO}}_{2}$ emissions during hard coal combustion), P-fertilizer (release of ${\mathrm{SO}}_{2}$ from land-filled gypsum, which is a by-product of TSP production), and hydrochloric acid (various ${\mathrm{SO}}_{2}$ sources along the supply chain).

EQ credits are given to the AD process primarily for the displacement of market biogas (reduced release of N-rich and P-rich compounds during digestion and sludge incineration). The spray-drying process profits from the displacement of market soy (reduction in pesticide and fertilizer use). The former yields an advantage for the AF + energy system in the subcategories of acidification, freshwater eutrophication, and terrestrial eutrophication. The latter yields an advantage for the AF + fodder system in the categories of marine eutrophication and ecotoxicity.

#### 3.1.3. Human Health

Both algal fuels show substantially higher impacts than petroleum diesel in five out of six Human Health (HH) subcategories. The only exception is ionizing radiation (HH rad) where they achieve an overall negative impact (environmental benefit) due to the displacement of nuclear electricity from the Spanish grid.

Impacts in the algal fuel pathways can be traced back to electricity consumed in the cultivation and milling processes (nuclear grid electricity), HCl consumption in the harvesting stage (electricity demand and Cl-gas and SOx emissions along the HCl supply chain), and the CHP supply chain (release of toxic substances during biowaste digestion and sludge incineration). Emissions from the wastewater treatment process (zinc, chromium VI, and NOx) are a product of the generic wastewater composition (cf. Section 2.2.1), and the actual HH impact of cultivation medium treatment might be lower than shown here.

AF + energy credits in the subcategories of carcinogenic effects (HH CE), non-carcinogenic effects (HH NCE), and ozone depletion (HH ODP) are driven by the displacement of market biogas, whereas fodder co-production yields no significant benefit. AF + energy performs favorably in the respiratory effects category (HH resp) for the same reason. AF + fodder shows lower impacts in the subcategories of ionizing radiation (HH rad) and photochemical ozone creation (HH POC), where it profits from the displacement of market soy (reduced slash and burn in Brazil) and from its higher net electricity production (cf. Section 2.2.2).

#### 3.1.4. Resource Depletion

When compared to petroleum diesel, the algal fuels perform unfavorably in three out of four resource depletion categories (res). In the fossil depletion category (res foss), the AF + energy system achieves significant impact reduction compared to petroleum diesel (credits for internal urea demand reduction and market biogas displacement), whereas AF + fodder is on par with petroleum diesel (impact from additional heat demand for spray-drying).

Water dissipation (res water) is driven by embedded impacts in the form of market biogas (biomass irrigation) and market urea (steam used as energy source and hydrogen source in ammonia production). As the AF + energy system consumes less of both, it has the lower impact. Note that seawater evaporation is not associated with a dissipation impact, as seawater is not a critical resource.

Both algae pathways have a similar land footprint (res land), which is dominated by the biogas supply chain (composting of biomass). The credit for market biogas substitution (AF + energy) is marginally bigger than the credit for soy meal substitution (AF + fodder), giving AF + energy a small advantage. Note that the land demand for algae cultivation itself is negligible in comparison.

Minerals and metal depletion (res MM) in both algal fuel systems is caused by the use of copper and zinc in buildings and appliances throughout various supply chains—most notably the production of HCl, biogas, fertilizer, and the treatment of wastewater. Soybean meal displacement in the AF + fodder pathway is rewarded a significant credit (displaced harvesting equipment and fertilizers), which is partially consumed by the additional energy demand of spray-drying. The AF + energy pathway, on the other hand, receives credits for the displacement of market biogas and the reduction in urea demand, and it is slightly more favorable.

### 3.2. Social Life Cycle Impacts

Figure 4 compares the social life cycle impact of the AF + energy and AF + fodder system (normalized with respect to the highest score between the two). Note that the indicator *sector contribution to economic development* is the only indicator expressed in medium *opportunity* hours (higher is better), whereas all other impact categories are expressed in medium *risk* hours (lower is better). Overall, the benefits of energy co-production, although relevant, appear less evident than those of fodder co-production. Whereas AF + energy features lower social risks in three out of nine categories (child labor, CL; sector contribution to the economic development, SED; women in the sectoral labor force, WLF), AF + fodder shows a favorable performance in the other six categories (fair salary, FS; forced labor, FL; gender wage gap, GWG; health expenditure, HE; unemployment, U; violations of employment laws and regulations, VEL).

Figure 5 shows a breakdown of risks according to location, distinguishing between Spain and the rest of the world (RoW). For most indicators, both pathways yield social benefits in the rest of the world but create burdens in Spain. The opposite is true for the SED category, which indicates that economic development is fostered domestically, while it is hampered internationally. Furthermore, the AF + fodder pathway augments social risks related to child labor and the share of women in the labor force in the rest of the world. Note that fuel conversion in the Netherlands was found to contribute less than 5% of the medium risk hours in all of the selected social life cycle categories.

Table 3 further refines the breakdown by listing the main risk and benefit drivers in each category. The risk side is dominated by Spanish domestic biogas and chemicals production. Although risk levels in these sectors are relatively low, they result in a relatively high absolute risk when multiplied by the high input demand per unit of algal fuel (cf. Table 1 and Table 2). The benefit side is led by the co-production of energy and fodder, which displace imports from hotspot countries, such as Russia, Nigeria, and Brazil. Again, the SED category shows the opposite trend, indicating that social risks in these countries are aggravated if economic opportunities are reduced. We recommend paying close attention to this trade-off in a potential decision-making context.

Our results mirror those from existing literature in several areas. By analyzing renewable hydrogen production in Spain instead of microalgal fuel production, Valente et al. [26] and Valente et al. [41] identified the same hotspot countries in the categories of CL, GWG, and HE. Tavakoli and Barkdoll [27] confirm our finding that the majority of the occupational benefits of algal fuel production are domestic, despite having a different scope (USA) and applying a self-developed SLCA methodology. The impact categories of FS, FL, U, VEL, and WLF cannot be compared because none of the cited studies address them.

### 3.3. Limitations

The models used in this study are subject to limitations, which should be accounted for when interpreting the presented ELCA and SLCA results.

We assume that algae can be grown in open ponds without applying pesticides. If pesticide use is necessary, the ecotoxicity impact of algal biomass production would be greater than shown in our study.

Although power demand for raceway pond operation is frequently reported, values found in the literature vary by orders of magnitude. Furthermore, the correlation between mixing power and biomass yield is rarely explored. As the two presented systems are net electricity exporters, they benefit from the displacement of grid electricity. Thus, impacts would increase if grid electricity is greener than modeled or if power demand increases (i.e., less power can be exported). Such changes would mainly affect the impact categories of climate change, ionizing radiation, and fossil resource depletion.

Our anaerobic digestion model does not account for the release of nitrogen-rich or phosphorus-rich compounds to water and air. Although digestate recycling should reduce this risk, our study may have underestimated it. Impacts in the ecosystem quality and human health categories would be exacerbated by the release of these compounds.

We assume that digestate can be recycled wholly and infinitely without impacting biomass yields—a practice which has yet to be proven at scale. If algae toxins are found to accumulate in the digestate, the recycled ratio would need to be reduced. In turn, fertilizer demand would increase, and an alternative digestate disposal route would have to be determined. This would likely increase impacts in all categories.

The treatment of discharged cultivation medium presents a significant source of environmental impacts in our study. To the authors’ best knowledge, no public data on the composition of spent algae cultivation media exists. Thus, we had to rely on a generic model from the Ecoinvent database to model its treatment. It is of particular concern that impacts in the human health category could be significantly lower than presented in this study. We recommend that this knowledge gap should be closed in future studies.

Our models rely on environmental and socio-economic background data, which are highly specific to the geographical and temporal scopes of this study. The obtained results should neither be applied to other countries nor be extended into the long-term future.

## 4. Conclusions

The presented study compares the potential environmental and social life cycle performance of microalgal fuel and fodder co-production (AF + fodder) against microalgal fuel and energy co-production (AF + energy) in Spain. Our environmental impact assessment shows a mixed picture, indicating that energy co-production outperforms fodder co-production in 11 out of 19 indicators. By contrast, the social risk assessment favors fodder co-production in six out of nine categories. We conclude that there is no systematic environmental or social benefit of fodder co-production over energetic utilization of the oil-extracted biomass. Preference for either option can only be established by weighing the environmental and social issues, which is inherently value-based and not further investigated. Our comparison of algal fuel to petroleum diesel further identifies the need for improvement in several environmental impact categories. Potential improvements include the use of wastewater as a nutrient source for algae cultivation and the optimization of operating conditions to minimize the residual concentrations of carbon-rich, nitrogen-rich, and phosphorous-rich compounds in the cultivation medium at the time of harvest. We further highlight the lack of publicly available data on the composition of cultivation effluent. Generally, our results show that co-production strategies have a decisive impact on the environmental and social performance of algal fuel. Hence, we recommend exploring new technologies and system configurations that enable truly sustainable algal fuel production.

## Figures and Tables

**Figure 1 ijerph-18-11351-f001:**
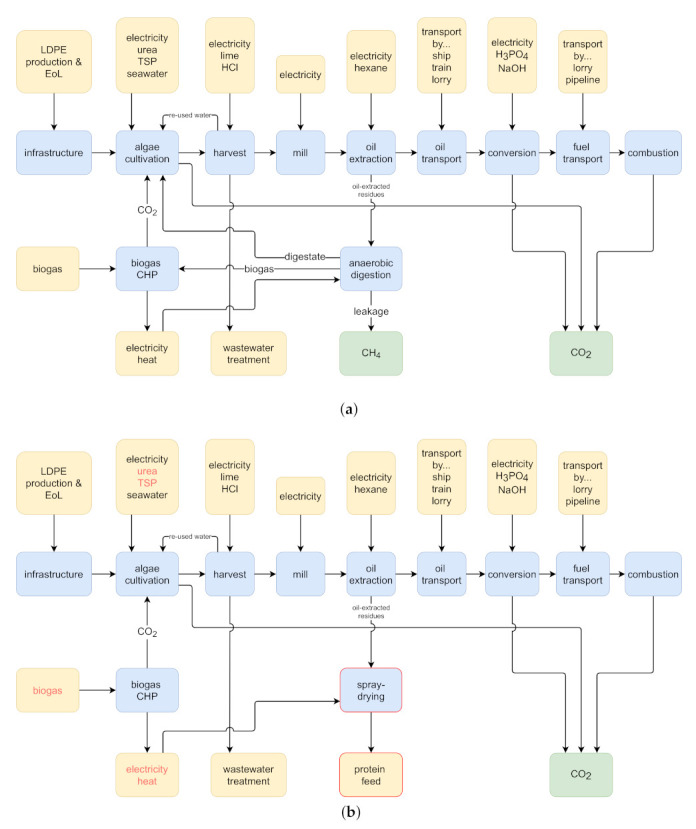
Scheme of the two studied algal fuel co-production systems. Yellow—background system; green—biosphere; blue—foreground system; red frame—process newly introduced in AF + fodder system; red font—identical process in both systems but with different flow amount. Abbreviations: LDPE—low density polyethylene; EoL—end of life; TSP—triple superphosphate; CHP—combined heat and power plant. (**a**) AF + energy system (adapted from [31]). (**b**) AF + fodder system.

**Figure 2 ijerph-18-11351-f002:**
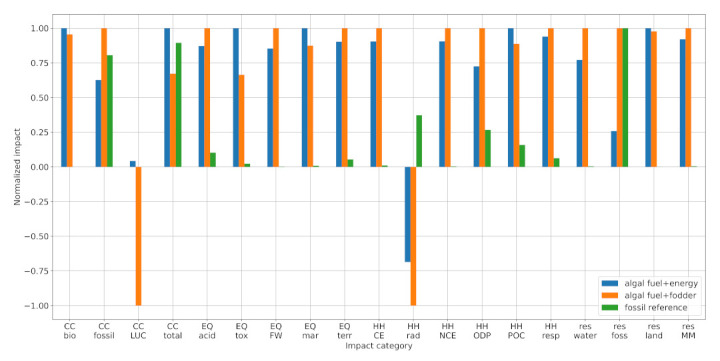
Comparison of environmental impacts: AF + energy (blue), AF + fodder (orange), and petroleum diesel (green). Note that the bars are normalized by the maximum in each category. Abbreviations: CC—climate change; EQ—ecosystem quality; HH—human health; res—resource depletion; bio—biogenic; LUC—land use and land use change; acid—freshwater and terrestrial acidification; tox—freshwater ecotoxicity; FW—freshwater eutrophication; mar—marine eutrophication; terr—terrestrial eutrophication; CE—carcinogenic effects; rad—ionizing radiation; NCE—non-carcinogenic effects; ODP—ozone depletion potential; POC—photochemical ozone creation; resp—respiratory effects; water—water dissipation; foss—fossil; land—land use; MM—minerals and metals.

**Figure 3 ijerph-18-11351-f003:**
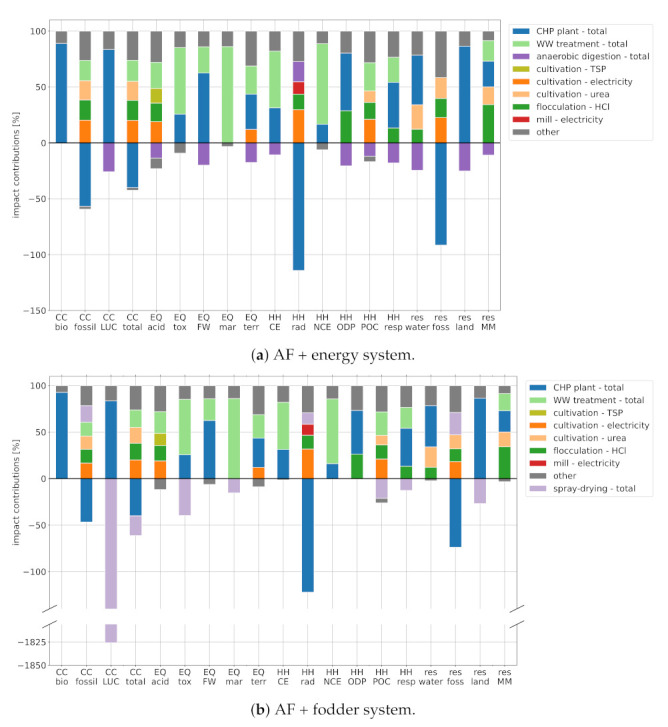
Break-down of environmental impacts: (**a**) AF + energy. (**b**) AF + fodder. Note that the bars are normalized by the sum of positive (i.e., damaging) impacts. Abbreviations: CC—climate change; EQ—ecosystem quality; HH—human health; res—resource depletion; bio—biogenic; LUC—land use and land use change; acid—freshwater and terrestrial acidification; tox—freshwater ecotoxicity; FW—freshwater eutrophication; mar—marine eutrophication; terr—terrestrial eutrophication; CE—carcinogenic effects; rad—ionizing radiation; NCE—non-carcinogenic effects; ODP—ozone depletion potential; POC—photochemical ozone creation; resp—respiratory effects; water—water dissipation; foss—fossil; land—land use; MM—minerals and metals.

**Figure 4 ijerph-18-11351-f004:**
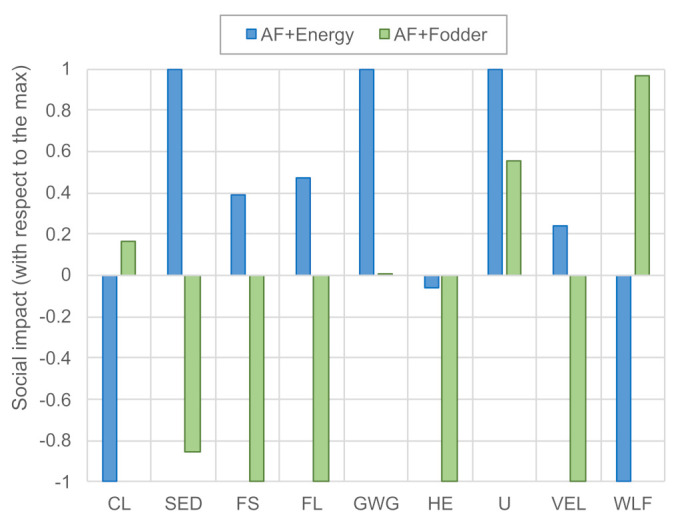
Social life cycle impacts of the AF + energy and AF + fodder systems. Scores are normalized by the highest absolute score in each impact category. Abbreviations: CL—Child Labor; SED—Sector contribution to Economic Development; FS—Fair Salary; FL—frequency of Forced Labor; GWG—Gender Wage Gap; HE—Health Expenditure; U—Unemployment; VEL—Violations of Employment Laws and regulations; WLF—Women in the sectoral Labor Force. Note that SED is the only positive indicator (higher is better), and all other indicators should be interpreted as “lower is better”.

**Figure 5 ijerph-18-11351-f005:**
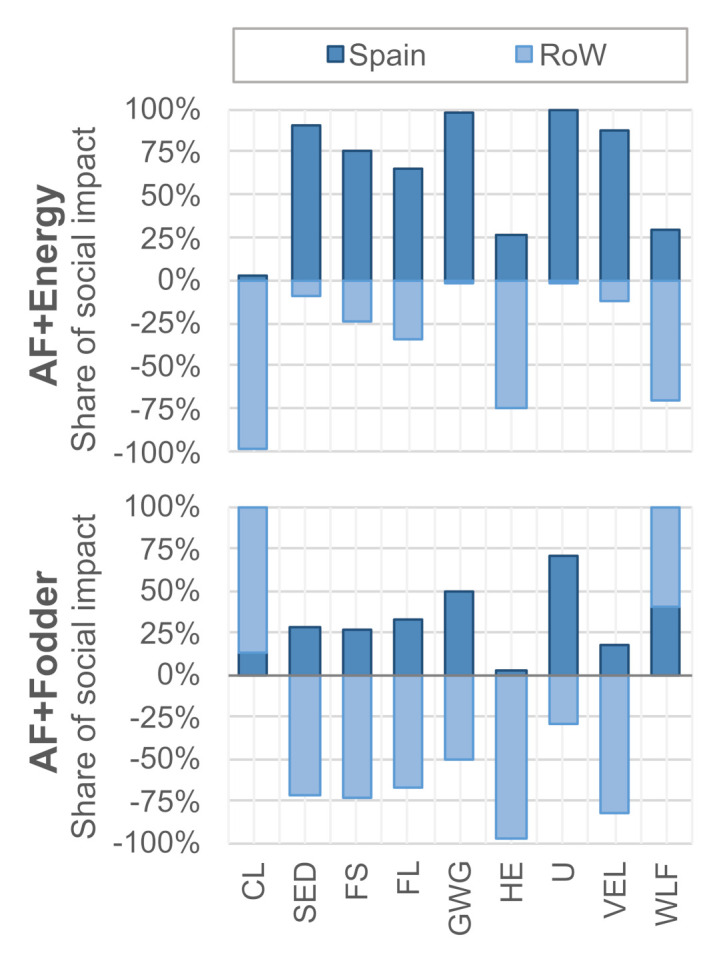
Breakdown of impact origins by domestic (Spain) and foreign (Rest of the World, RoW) activities. Abbreviations: CL—Child Labor; SED—Sector contribution to Economic Development; FS—Fair Salary; FL—frequency of Forced Labor; GWG—Gender Wage Gap; HE—Health Expenditure; U—Unemployment; VEL—Violations of Employment Laws and regulations; WLF—Women in the sectoral Labor Force. Note that SED is the only positive indicator (higher is better), and all other indicators should be interpreted as “lower is better”.

**Table 1 ijerph-18-11351-t001:** Inventory of the AF + energy scenario (adapted from Portner et al. [31]).

Material	Ecoinvent 3.7.1 APOS Activity, Location (ELCA) ^1^	PSILCA 3.0 Sector, Country (SLCA)	Amount per MJ	Unit	Cost per FU (USD)
Biogas	market for biogas, RoW	Agricultural and livestock services, ES	1.23 × 10${}^{-1}$	m³	1.42 × 10${}^{-2}$
N-fertilizer	nutrient supply from urea, RER	Manufacture of pesticides and agrochemical products, ES	5.18 × 10${}^{-3}$	kg N	2.89 × 10${}^{-3}$
P-fertilizer	nutrient supply from triple superphosphate, RER	Manufacture of pesticides and agrochemical products, ES	1.04 × 10${}^{-3}$	kg P₂O₅	2.96 × 10${}^{-4}$
Pond liner	market for packaging film, low density polyethylene, GLO	Manufacture of plastic products, ES	5.46 × 10${}^{-4}$	kg	1.91 × 10${}^{-3}$
Water pipeline	market for water supply network, GLO	Civil Engineering, ES	5.07 × 10${}^{-}9$	km	5.10 × 10${}^{-3}$
Lime	market for lime, RER	Basic chemical products, ES	4.91 × 10${}^{-2}$	kg	6.78 × 10${}^{-3}$
HCl	market for hydrochloric acid, without water, in 30% solution state, RER	Basic chemical products, ES	4.83 × 10${}^{-2}$	kg (undiluted)	7.24 × 10${}^{-3}$
Hexane	market for hexane, GLO	Coke, refined petroleum products and nuclear fuel, ES	1.64 × 10${}^{-4}$	kg	6.31 × 10${}^{-5}$
Rail transport	market group for transport, freight train, RER	Railway transport, ES	3.16 × 10${}^{-3}$	t km	9.68 × 10${}^{-5}$
Sea transport	market for transport, freight, sea, tanker for petroleum, GLO	Water transport, ES	1.11 × 10${}^{-1}$	t km	4.13 × 10${}^{-5}$
Road transport	market for transport, freight, lorry 16–32 metric ton, EURO6, RER	Other transport material n.e.c., ES	2.74 × 10${}^{-3}$	t km	9.34 × 10${}^{-5}$
Pipeline transport	market for transport, pipeline, onshore, petroleum, RER	Other land transport; transport via pipelines, ES	9.25 × 10${}^{-3}$	t km	5.17 × 10${}^{-5}$
Electricity	market for electricity, medium voltage, NL	Electricity, gas, steam and hot water supply, NL	2.11 × 10${}^{-3}$	kWh	2.49 × 10${}^{-4}$
H₃PO₄	market for phosphoric acid, industrial grade, without water, in 85% solution state, GLO	Manufacture of industrial chemicals and fertilizers, IL	1.95 × 10${}^{-5}$	kg (undiluted)	2.07 × 10${}^{-5}$
NaOH	market for sodium hydroxide, without water, in 50% solution state, GLO	Manufacture of chemicals and chemical product, NL	5.84 × 10${}^{-5}$	kg (undiluted)	1.32 × 10${}^{-5}$
Electricity (avoided)	market for electricity, high voltage, ES	Production and distribution of electricity, ES	−1.45 × 10${}^{-1}$	kWh	−1.68 × 10${}^{-2}$
Natural gas (avoided)	heat and power co-generation, natural gas, combined cycle power plant, 400MW electrical, ES	Natural gas mix, ES ${}^{2}$	−1.30	MJ	−1.63 × 10${}^{-2}$

${}^{1}$ Where no local Ecoinvent activity was available, the next-largest parent region was chosen (RER, Europe w/o Switzerland, GLO). If only CH and RoW were available, CH was preferred. ${}^{2}$ Spanish natural gas mix: Algeria 33.11%; Nigeria 11.47%; Qatar 11.46%; US 11.04%; Russia 8.52%; Trinidad and Tobago 7.52%; France 7.02%; Norway 6.52%; Peru 1.43%; Angola 0.73%; Portugal, 0.25%; Cameroon 0.23% (source: www.ine.es).

**Table 2 ijerph-18-11351-t002:** Inventory of the AF + fodder scenario.

Material	Ecoinvent 3.7.1 APOS Activity, Location (ELCA) ^1^	PSILCA 3.0 Sector, Country (SLCA)	Amount per MJ	Unit	Cost per FU (USD)
Biogas	market for biogas, RoW	Agricultural and livestock services, ES	1.72 × 10${}^{-1}$	m³	2.22 × 10${}^{-2}$
N-fertilizer	nutrient supply from urea, RER	Manufacture of pesticides and agrochemical products, ES	9.44 × 10${}^{-3}$	kg N	5.27 × 10${}^{-3}$
P-fertilizer	nutrient supply from triple superphosphate, RER	Manufacture of pesticides and agrochemical products, ES	1.90 × 10${}^{-3}$	kg P₂O₅	5.39 × 10${}^{-4}$
Pond liner	market for packaging film, low density polyethylene, GLO	Manufacture of plastic products, ES	5.46 × 10${}^{-4}$	kg	1.91 × 10${}^{-3}$
Water pipeline	market for water supply network, GLO	Civil Engineering, ES	5.07 × 10${}^{-}9$	km	5.10 × 10${}^{-3}$
Lime	market for lime, RER	Basic chemical products, ES	4.91 × 10${}^{-2}$	kg	6.70 × 10${}^{-3}$
HCl	market for hydrochloric acid, without water, in 30% solution state, RER	Basic chemical products, ES	4.83 × 10${}^{-2}$	kg (undiluted)	7.24 × 10${}^{-3}$
Hexane	market for hexane, GLO	Coke, refined petroleum products and nuclear fuel, ES	1.64 × 10${}^{-4}$	kg	6.31 × 10${}^{-5}$
Rail transport	market group for transport, freight train, RER	Railway transport, ES	3.16 × 10${}^{-3}$	t km	9.68 × 10${}^{-5}$
Sea transport	market for transport, freight, sea, tanker for petroleum, GLO	Water transport, ES	1.11 × 10${}^{-1}$	t km	4.13 × 10${}^{-5}$
Road transport	market for transport, freight, lorry 16–32 metric ton, EURO6, RER	Other transport material n.e.c., ES	2.74 × 10${}^{-3}$	t km	9.34 × 10${}^{-5}$
Pipeline transport	market for transport, pipeline, onshore, petroleum, RER	Other land transport; transport via pipelines, ES	9.25 × 10${}^{-3}$	t km	5.17 × 10${}^{-5}$
Electricity	market for electricity, medium voltage, NL	Electricity, gas, steam and hot water supply, NL	2.11 × 10${}^{-3}$	kWh	2.49 × 10${}^{-4}$
H₃PO₄	market for phosphoric acid, industrial grade, without water, in 85% solution state, GLO	Manufacture of industrial chemicals and fertilizers, IL	1.95 × 10${}^{-5}$	kg (undiluted)	2.07 × 10${}^{-5}$
NaOH	market for sodium hydroxide, without water, in 50% solution state, GLO	Manufacture of chemicals and chemical product, NL	5.84 × 10${}^{-5}$	kg (undiluted)	1.32 × 10${}^{-5}$
Heat	heat and power co-generation, natural gas, combined cycle power plant, 400 MW electrical, ES	Natural gas mix, ES ^2^	6.10 × 10${}^{-1}$	MJ	7.63 × 10${}^{-3}$
Tap water	market for tap water, Europe w/o Switzerland	Collection, purification and distribution of water, ES	4.91 × 10${}^{-2}$	kg	2.30 × 10${}^{-5}$
Electricity (avoided)	market for electricity, high voltage, ES	Production and distribution of electricity, ES	−1.74 × 10${}^{-1}$	kWh	−2.01 × 10${}^{-2}$
Soybean (avoided)	soybean meal and crude oil production, BR	Processed soy oil, BR	−3.85 × 10${}^{-2}$	kg	−2.40 × 10${}^{-2}$
Soybean (avoided)	soybean meal and crude oil production, US	Soybean and other oilseed processing, US	−3.15 × 10${}^{-2}$	kg	−1.96 × 10${}^{-2}$

${}^{1}$ Where no local Ecoinvent activity was available, the next-largest parent region was chosen (RER, Europe w/o Switzerland, GLO). If only CH and RoW were available, CH was preferred. ${}^{2}$ Spanish natural gas mix: Algeria 33.11%; Nigeria 11.47%; Qatar 11.46%; US 11.04%; Russia 8.52%; Trinidad and Tobago 7.52%; France 7.02%; Norway 6.52%; Peru 1.43%; Angola 0.73%; Portugal, 0.25%; Cameroon 0.23% (source: www.ine.es).

**Table 3 ijerph-18-11351-t003:** Summary of social risk drivers.

Social Indicator	Main Benefit Driver AF + Energy	Main Impact Driver AF + Energy	Main Benefit Driver AF + Fodder	Main Impact Driver AF + Fodder
Child labor, total	Displacement of natural gas from Nigeria and Russia	Chemicals production in Spain	(i) Displacement of soy from Brazil; (ii) Displacement of Spanish grid electricity	(i) Natural gas production in Nigeria and Russia; (ii) Chemicals production in Spain
Sector contribution to economic development (positive indicator)	Algal fuel production in Spain	Displacement of energy products from Nigeria, Russia, and South Africa	Production of biogas and chemicals in Spain	Displacement of Brazilian soy
Fair Salary	Displacement of activities in Spain, Algeria related to energy products	Production of biogas and chemicals in Spain	Displacement of soy from Brazil and USA	Production of biogas and chemicals in Spain
Frequency of forced labor	Displacement of activities in Spain, Algeria, and Russia related to energy products	Production of biogas and chemicals in Spain	Displacement of Brazilian soy	(i) Natural gas production in Nigeria, Algeria, and Russia; (ii) Production of biogas and chemicals in Spain
Gender wage gap	Displacement of Spanish grid electricity	Chemicals production in Spain	(i) Displacement of soy from Brazil and USA (ii) Displacement of Spanish grid electricity	(i) Production of biogas and chemicals in Spain; (ii) Natural gas production in Peru
Health expenditure	Displacement of natural gas from Nigeria, Cameroon, and Qatar	Chemicals production in Spain	Displacement of Brazilian soy	Algal fuel production in Spain
Unemployment	Displacement of Spanish grid electricity	Production of biogas, chemicals, and water in Spain	(i) Displacement of Spanish grid electricity; (ii) Displacement of Brazilian soy	Production of biogas and chemicals in Spain
Violations of employment laws and regulations	Displacement of natural gas from USA and Peru	Production of biogas, chemicals, and water infrastructure in Spain	Displacement of US soy	Production of biogas and chemicals in Spain
Women in the sectoral labor force	Displacement of economic activities in France, Peru, and Algeria related to energy products	Production of biogas and chemicals in Spain	Displacement of Spanish grid electricity	(i) Production of biogas, chemicals, and water infrastructure in Spain; (ii) Natural gas-related activities in France

## Data Availability

The AF + fodder model used in this studies is available in the supporting information under https://www.mdpi.com/article/10.3390/ijerph182111351/s1. The Af + energy model is available in the supporting information of Portner et al. [31]. The latter has been modified for this study as described in https://www.mdpi.com/article/10.3390/ijerph182111351/s2.

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
