# Peer review of "Sustainability Assessment of Combined Animal Fodder and Fuel Production from Microalgal Biomass"

_ijerph, 2021, doi:10.3390/ijerph182111351_

Round 1

Reviewer 1 Report

This study presented a comparative environmental and social life cycle assessment of algal fuel and fodder co-production versus algal fuel and energy co-production 3 (AF+energy). It’s interesting but it still needs clarification. I recommend a major revision.

  1. The reason to do the research is not well explained in the introduction.
  2. The advantages of LCA and the reason to use it are not well explained as well.
  3. The significance of the research has not been clarified.
  4. Since ELCA and SLCA are two models, how could the author do validation for the models?
  5. Line 224, what does “total” mean?
  6. Could the author provide how they calculate each parameter?
  7. Could the author provide the data sources? How many references? Where and how do they find them?
  8. The conclusion needs to be extended. It looks a little weak. What are the main discoveries?
  9. What is the environmental implication of the study?

Reviewer 2 Report

This is an interesting area and the benefits of using algal fuel by-products as fodder (animal food) rather than energy (heat and electricity) production. The article demonstrates the complexity of the issue across a comprehensive set of environmental impacts, and concludes that there are advantages to both uses of algal fuel by-products.

Further comment on the applicability of these findings to countries beyond Spain would be helpful for the reader in assessing the global impact of these findings. Whilst this has been considered in the social impacts, it is obvious that many of the resources required for these processes could vary in cost and environmental impact in different parts of the world. Since algal fuel appears to be outperformed by more traditional fossil fuels in many of these areas, it would be interesting to identify areas for improvement that might allow algal fuel to become a competitor in this sense. The findings seem to somewhat overshadow the differences between AF+fodder and AF+energy, so further discussion is required.

The content of this article is very interesting and relevant, as well as being a good fit for the journal, so I highly recommend this for publication in International Journal of Environmental Research and Public Health.

Reviewer 3 Report

The research article titled “Sustainability Assessment of Combined Animal Fodder and Fuel Production from Microalgal Biomass” aimed to evaluate a comparative environmental and social life cycle assessment (ELCA and SLCA) of algal fuel and fodder co-production (AF+fodder) versus algal fuel and energy co-production (AF+energy). However, none of the compared systems were found to yield a consistent advantage in the environmental or social dimension and therefore it’s not possible to recommend a co-production strategy without weighting environmental and social issues.

However, the article cannot be accepted in its current form as it offers limited value in terms of new insightfulness on scientific basis. Although the authors have conducted an extensive study accounting comprehensive indicators for environmental and social life cycle assessment, the results are presented in very broad spectrum which makes it extremely difficult to comprehend and apply these results for designing a sustainable bioprocess based on algal biorefinery. The authors need to refine the manuscript to enable easy following of the logical trail of thought and offer clear insights into plausible ways for advancements in algal fuel sustainability.

Further, some of the other pertinent points which need to be addressed are pointed below:

  1. The authors can avoid writing in first person throughout the manuscript.
  2. The discussion part needs to be thoroughly revised in the entire manuscript for clear understanding of the implications of observed results.
  3. The authors are also suggested to discuss the obtained results in corroboration with similar studies, so that despite accounting for variations a breakthrough can be envisaged for advancement in algal technology for both value-added or energy production.
  4. The repetition of information such as “Our environmental impact assessment shows a mixed picture, indicating that energy co-production outperforms fodder co-production in 11 out of 19 indicators” can be avoided in conclusion.
  5. For Fig. 5, For most indicators, both pathways yield social benefits in the rest of the world but create burdens in Spain. Kindly raise a rationale for this disparity.

Round 2

Reviewer 1 Report

It has been improved a lot, and could be published. 

Reviewer 3 Report

The authors have justified the comments made during the previous review.